# ACTIVITY REGULARIZATION FOR CONTINUAL LEARNING

## ABSTRACT

While deep neural networks have achieved remarkable successes, they suffer the well-known *catastrophic forgetting* issue when switching from existing tasks to tackle a new one. In this paper, we study *continual learning* with deep neural networks that learn from tasks arriving sequentially. We first propose an approximated multi-task learning framework that unifies a family of popular regularization based continual learning methods. We then analyze the weakness of existing approaches, and propose a novel regularization method named "Activity Regularization" (AR), which alleviates forgetting meanwhile keeping model's plasticity to acquire new knowledge. Extensive experiments show that our method outperforms state-of-the-art methods and effectively overcomes catastrophic forgetting.

## 1 INTRODUCTION

Human has the ability to continuously learn, accumulate knowledge in their lifetime and be able to retrieve, consolidate previously acquired skills whenever necessary. Such kind of ability is referred as *continual learning*, which is a fundamental capability contributing to the skills development and specialization in humankind. Conventional machine learning, especially neural networks, however, are not capable of learning continuously from solving one task to another without forgetting the previous knowledge. This phenomenon is known as the *catastrophic forgetting* or *catastrophic inference* which has been a long-standing challenge in machine learning and AI for years. To overcome the catastrophic forgetting problem, a deep neural network on one hand should not be allowed to change too drastically in order to avoid forgetting the previous knowledge; on the other hand, the same network also should be flexible enough to learn and acquire new knowledge. Such condition is referred to as the *stability-plasticity dilemma* (Abraham & Robins, 2005) which has posed a great challenge in continual learning. In literature, continual learning has been extensively studied in many fields, ranging from robotics, machine learning, to neuroscience and cognitive science.

In this work, we investigate the problem of continual learning from machine learning perspective. In particular, we show that catastrophic forgetting can be alleviated by optimizing the multi-task learning objective in which popular approaches such as Elastic Weight Consolidation (EWC) (Kirkpatrick et al., 2017), Synaptic Intelligence (SI) (Zenke et al., 2017) and Gradient Episodic Memory (Lopez-Paz et al., 2017) (GEM) can be viewed as special cases. We show that by varying the losses in the approximated multi-task learning objective, we can recover a family of regularization based approaches for continual learning. Then, we analyze the weakness of two popular approaches, EWC and GEM, and show that label noise or mistakes in the memory can result in dramatic changes in the optimal decision boundary of previous tasks, leading to catastrophic forgetting.

To overcome the above problem, we propose a novel technique called "Activity Regularization" (AR) that penalizes changes in the model's prediction on its learned information. AR works by utilizing a memory to store some of previous training samples; Then, when learning a new task, AR penalizes the KL-divergence between the current model's and the previous optimal model's predictions on the corresponding memory. This objective is to ensure that the new model will be consistent with the previous optimal ones on the old tasks, thus avoiding catastrophic forgetting. We further develop a stochastic version of AR that randomly samples a task and apply AR in each mini-batch update. Stochastic AR allows the network to be more flexible when acquiring new knowledge while still alleviating catastrophic forgetting. We evaluate our method on popular continual learning benchmarks and achieve promising results compared with state-of-the-art methods.

## 2   CONTINUAL LEARNING BY APPROXIMATED MULTI-TASK LEARNING

**Basic notations.** We first present notations used in this work. We use subscripts to denote task index, for example, $\mathcal{T}_1$, $\mathcal{D}_1$, are the first task and first training data respectively. Superscripts denote sample in a data set, we omit subscripts of the samples if the task is known, for example, $\mathcal{D}_t = \{\boldsymbol{x}^{(n)}, y^{(n)}\}_{n=1}^N$ means data set $\mathcal{D}_t$ consists of $n$ training samples. We denote a model that is trained on task $t$ and the current model as $\boldsymbol{\theta}_t$ and $\boldsymbol{\theta}$ respectively. Finally, $\boldsymbol{\theta}^k$ denotes the $k$–th parameter of $\boldsymbol{\theta}$.

Consider the continual learning problem where the learner $\boldsymbol{\theta}$ receives a sequence of T tasks $\mathcal{T} = \{\mathcal{T}_1, \mathcal{T}_2, \ldots, \mathcal{T}_T\}$, each of which consists of a training data $\mathcal{D}_t$. At any given time $t$, only task $\mathcal{T}_t$ are presented to the learner $\boldsymbol{\theta}$ and it has to learn to solve the current task without access to the training data of any previous tasks. The goal of continual learning is to train the learner such that it can solve the not only current task $\mathcal{T}_t$ but also all the previous tasks $\mathcal{T}_{k<t}$.

To avoid forgetting previous knowledge, when learning task $\mathcal{T}_t$, the learner $\boldsymbol{\theta}$ minimize the multi-task learning objective as

$$\tilde{\mathcal{L}}_t(\boldsymbol{\theta}) = \mathcal{L}_t(\boldsymbol{\theta}, \mathcal{D}_t) + \lambda_{t-1}\mathcal{L}_{t-1}(\boldsymbol{\theta}, \mathcal{D}_{t-1}) + \ldots + \lambda_1\mathcal{L}_1(\boldsymbol{\theta}, \mathcal{D}_1), \tag{1}$$

where the loss $\mathcal{L}_i(\boldsymbol{\theta}, \mathcal{D}_i)$ is usually the negative log-likelihood $-\sum_{\boldsymbol{x},y \in \mathcal{D}_i} \log p(y|\boldsymbol{x}; \boldsymbol{\theta})$ of the corresponding training data and $\lambda_i$ regulates the importance of task $\mathcal{T}_i$ to all the tasks observed so far. In continual learning, at time $t$, we do not have access to the data $\mathcal{D}_{k<t}$ to evaluate the loss $\mathcal{L}_{k<t}$, thus, we need to approximate the past losses $\mathcal{L}_{k<t}(\boldsymbol{\theta}, \mathcal{D}_k)$ when minimizing $\mathcal{L}_t(\boldsymbol{\theta}, \mathcal{D}_t)$.

A common choice is to employ Laplace propagation (Eskin et al., 2004) of the negative log posterior

$$-\log p(\boldsymbol{\theta}|\mathcal{D}_i) \approx -\log p(\boldsymbol{\theta}_i^*|\mathcal{D}_i) + \frac{1}{2}(\boldsymbol{\theta} - \boldsymbol{\theta}_i^*)^T \boldsymbol{F}_i (\boldsymbol{\theta} - \boldsymbol{\theta}_i^*), \tag{2}$$

where $\boldsymbol{\theta}_i^*$ is a mode of the log posterior and $\boldsymbol{F}_i$ is the empirical Fisher information matrix evaluated at $\boldsymbol{\theta} = \boldsymbol{\theta}_i^*$. If we assume an uniform prior on $\boldsymbol{\theta}$, the posterior coined with the likelihood [1], thus by substituting eqn. 2 to 1, the objective for training task $\mathcal{T}_t$ becomes

$$\tilde{\mathcal{L}}_t(\boldsymbol{\theta}) = \mathcal{L}_t(\boldsymbol{\theta}, \mathcal{D}_t) + \sum_{i<t} \lambda_k (\boldsymbol{\theta} - \boldsymbol{\theta}_i^*)^T \boldsymbol{F}_i (\boldsymbol{\theta} - \boldsymbol{\theta}_i^*), \tag{3}$$

where the terms $-\log p(\boldsymbol{\theta}_i^*|\mathcal{D}_i)$ are omitted because they are constant with respect to $\boldsymbol{\theta}$. According to eqn. 3, we can minimize the negative log-likelihood of $\boldsymbol{\theta}$ on the previous tasks without explicitly storing the training samples because the data distribution are already captured in the Fisher information. Eqn. 3 results in a wide range of methods for continual learning, each of which proposes a different method to approximate the Fisher information matrix. For example, Elastic Weight Consolidation (EWC) (Kirkpatrick et al., 2017) assumes the Fisher is diagonal and approximates it using the identity:

$$\boldsymbol{F}_t \approx \text{diag}\left(\sum_{n=1}^N \left(\nabla_{\boldsymbol{\theta}} \log p(\boldsymbol{y}_t^{(n)}|\boldsymbol{x}_t^{(n)}; \boldsymbol{\theta})\right)^2 \Big|_{\boldsymbol{\theta} = \boldsymbol{\theta}_t^*}\right) \tag{4}$$

Instead of using the Fisher's identity, Synaptic Intelligence (SI) (Zenke et al., 2017) directly measures the sensitivity of a parameter $\boldsymbol{\theta}_t^k$ to the loss function at task $\mathcal{T}_t$ through out the whole training trajectory. Zenke et al. (2017) also shows that this estimation is equivalent to measuring the full Fisher matrix on whole data set under certain choices of the loss function.

Another approach to achieve the approximated multi-task learning objective in eqn. 1 is based on the observation that at the beginning of task $\mathcal{T}_t$, we initialize $\boldsymbol{\theta} = \boldsymbol{\theta}_{t-1}^*$, whose value already minimized the lost $\tilde{\mathcal{L}}_{t-1}$. From this, if we can consider eqn. 1 as minimizing the loss $\mathcal{L}_t(\boldsymbol{\theta})$ with the constraints that previous losses are not allowed to increase after every mini-batch update, which is the GEM objective. In particular, GEM optimize the following problem

$$\begin{aligned} \text{Minimize}_{\boldsymbol{\theta}, \mathcal{D}_t} \quad & \mathcal{L}_t(\boldsymbol{\theta}) \\ \text{subject to} \quad & \mathcal{L}_k(\boldsymbol{\theta}, \mathcal{M}_k) \leq \mathcal{L}_k(\boldsymbol{\theta}_{t-1}^*, \mathcal{M}_k), \forall k < t, \end{aligned} \tag{5}$$

---

[1]With constant prior: $p(\boldsymbol{\theta}) = \text{const}, \forall \boldsymbol{\theta}$, by Bayes's rule: $p(\boldsymbol{\theta}|\mathcal{D}_i) \propto p(\boldsymbol{\theta})p(\mathcal{D}_i|\boldsymbol{\theta}) \propto p(\mathcal{D}_i|\boldsymbol{\theta})$.

where the loss $\mathcal{L}_k$ is evaluated at the memory $\mathcal{M}_k$ that stores some of the training samples from task $\mathcal{T}_k$. GEM further converts the constraints in eqn. 5 into the gradient constraints of the losses and solve them using quadratic programming.

Finally, the type of losses $\mathcal{L}_{k<t}$ can be flexibly changed to other loss rather than the negative log-likelihood. For example, Learning without Forgetting (LwF) (Li & Hoiem, 2017) proposes a form of distillation loss by minimizing the KL-divergence of past models and current model on the newly observed tasks. By changing the losses $\mathcal{L}_{k<t}$, we can recover different solutions for continual learning in the literature.

## 3 ACTIVITY REGULARIZATION FOR CONTINUAL LEARNING

### 3.1 LIKELIHOOD MATCHING IN APPROXIMATED MULTI-TASK LEARNING

While EWC and GEM are two popular methods for continual learning, we argue that they are sensitive to misclassified samples selected in the memory or used to estimate the Fisher information.

Let us consider a scenario of learning two tasks $\mathcal{T}_1$ and $\mathcal{T}_2$ where we store the first optimal model $\boldsymbol{\theta}_1^*$ and use a sample $\{\boldsymbol{x}_1, \boldsymbol{y}_1\}$ to estimate the Fisher information or storing in the memory. For EWC, by applying eqn. 2 with this scenario, we have:

$$(\boldsymbol{\theta} - \boldsymbol{\theta}_1^*)^T \boldsymbol{F}_1 (\boldsymbol{\theta} - \boldsymbol{\theta}_1^*) \approx \log p(\boldsymbol{\theta}_1^*|\boldsymbol{x}_1) - \log p(\boldsymbol{\theta}|\boldsymbol{x}_1), \tag{6}$$

again we assume uniform prior on $\boldsymbol{\theta}$, thus minimizing LHS of eqn. 6 is equivalent to finding $\boldsymbol{\theta}$ that has the same log-likelihood as $\boldsymbol{\theta}_1^*$ on $\boldsymbol{x}_1$. For GEM, the constraint is to penalize whenever $\boldsymbol{\theta}$ that has higher negative log-likelihood than $\boldsymbol{\theta}_1^*$ on $\boldsymbol{x}_1$. Now assume $\boldsymbol{y}_1 = [1, 0, 0]$ and the predictions of the two models $\boldsymbol{\theta}_1^*$ and $\boldsymbol{\theta}$ are $[0.1, 0.9, 0.1]$ and $[0.1, 0.1, 0.9]$, that is, both models make mistakes on $\boldsymbol{x}_1$. As a result, $\boldsymbol{\theta}_1^*$ and $\boldsymbol{\theta}$ have the same log-likelihood value on $\boldsymbol{x}_1$, and thus there are no penalty on $\boldsymbol{\theta}$ in both EWC and GEM. However, these two models are very different from each other since they make different mistakes on $\boldsymbol{x}_1$. In general, if we have too many misclassified samples or label noise, the decision boundary on the previous task might be altered despite minimizing the EWC or GEM objective.

This phenomenon happens based on an assumption that we usually do not train the model to have 0 error on the training set to avoid overfitting and we only use a relatively small data portion to estimate the Fisher or store as memory. Therefore, EWC and GEM might not correctly penalize the current model from changing the past tasks' optimized decision boundaries.

### 3.2 ACTIVITY REGULARIZATION FOR CONTINUAL LEARNING

To tackle the aforementioned problem, we propose to minimize the KL-divergence between the current model and the previous optimal models on the corresponding tasks. This can also be interpreted as distilling the knowledge from $\boldsymbol{\theta}_i^*$ to $\boldsymbol{\theta}$ on the corresponding data $\mathcal{D}_i$.

In particular, for each task $\mathcal{T}_i$, we store some of the samples $\boldsymbol{x}_i$ and the prediction of the optimal model $f_{\boldsymbol{\theta}_i^*}(\boldsymbol{x}_i)$ in a memory $\mathcal{M}_i$; then the Activity Regularization (AR) objective for task $\mathcal{T}_t$ is

$$\tilde{\mathcal{L}}_t(\boldsymbol{\theta}) = \mathcal{L}_t(\boldsymbol{\theta}) + \sum_{i<t} \sum_{\boldsymbol{x} \in \mathcal{M}_i} \lambda_i KL\left(\frac{1}{\tau} f_{\boldsymbol{\theta}_i^*}(\boldsymbol{x}) \,\Big\|\, \frac{1}{\tau} f_{\boldsymbol{\theta}}(\boldsymbol{x})\right), \tag{7}$$

where $\tau$ is the temperature used in softmax outputs. Our goal is to ensure that the new model $\boldsymbol{\theta}$ behaves similarly with $\boldsymbol{\theta}_t^*$ on the previous task $\mathcal{T}_t$, thus we call this constraint as activity regularization. Algorithm 1 gives a summary of the proposed Activity Regularization based algorithm for continual learning. We refer to this algorithm as "Deterministic Activity Regularization" (DAR) which imposes regularization constraints for all the previous tasks at every learning iteration.

**Remark.** Similar to approximated multi-task learning methods, DAR employs a regularization constraint on every of the past tasks when learning a new one, which requires calculating the KL-divergence resulting in an additional forward pass on the memory, and thus results in additional computational cost when training the models. Moreover, the number of regularizers for each task is equal to the number of previously observed tasks, which may considerably restrict the model's *plasticity* to acquire new knowledge. That is, by enforcing too many constraints on the current model for reducing forgetting, we sacrifice its ability to learn new information.

---

**Algorithm 1** Deterministic Activity Regularization (DAR) algorithm for Continual Learning.

---

1: Initialize the model and memory: $\boldsymbol{\theta}$, $\mathcal{M}_0 \leftarrow \emptyset$, select temperature $\tau$
2: **for** t = 1,...,T **do**
3:     Observe the data set $\mathcal{D}_t$
4:     $\mathcal{M}_t \leftarrow$ add random samples from $\mathcal{D}_t$ to the memory.
5:     **for** k = 1,...,$n_{iter}$ **do**
6:         Sample $\mathcal{D} = \{\boldsymbol{x}^{(n)}, y^{(n)}\}_{n=1}^N$ a batch from training data $\mathcal{D}_t$

7: 
$$g_{\boldsymbol{\theta}} \leftarrow \nabla_{\boldsymbol{\theta}} \left[ \sum_{\boldsymbol{x},y\in\mathcal{D}} \mathcal{L}_t(\boldsymbol{\theta}, (\boldsymbol{x}, y)) + \sum_{i<t} \sum_{\boldsymbol{x}\in\mathcal{M}_i} \lambda_i KL\left( \frac{1}{\tau} f_{\boldsymbol{\theta}_i^*}(\boldsymbol{x}) \,\middle\|\, \frac{1}{\tau} f_{\boldsymbol{\theta}}(\boldsymbol{x}) \right) \right]$$

8:         $\boldsymbol{\theta} \leftarrow \boldsymbol{\theta} - \alpha \cdot \text{SGD}(\boldsymbol{\theta}, g_{\boldsymbol{\theta}})$
9: **return** $\boldsymbol{\theta}$

---

### 3.3 STOCHASTIC ACTIVITY REGULARIZATION

To address the weaknesses of DAR, we propose the Stochastic Activity Regularization (SAR), which randomly samples a past task and apply activity regularization on each gradient update when learning a new task. Algorithm 1 summarizes the proposed SAR algorithm for continual learning.

For each task, we randomly sample its training data to add to the memory $\mathcal{M}_t$. Then, when learning a new task, for each mini batch to update $\boldsymbol{\theta}$, we apply AR on all of the previous memory. For SAR, we instead randomly sample an index i of the previous tasks and apply the AR only on that memory, As a result, SAR is much more robust as it improves the model's plasticity, alleviates catastrophic forgetting and has greater scalability. For DAR, we omit the sampling index i in step 7 and calculate the KL divergence on all the memory as in eqn. 7.

The memory $\mathcal{M}_i$ is used to store some training samples of the previous tasks which is used to regularize the current model when learning new tasks. Using a memory for continual learning has been explored by previous works Nguyen et al. (2018) and Lopez-Paz et al. (2017). Although more sophisticated methods such as the greedy $K$-center algorithm can be used to select data points that spread through out the input space (Nguyen et al., 2018), in this work, even random sampling works well with our methods and can achieve state-of-the-art results.

---

**Algorithm 2** Stochastic Activity Regularization (SAR) algorithm for Continual Learning.

---

1: Initialize the model and memory: $\boldsymbol{\theta}$, $\mathcal{M}_0 \leftarrow \emptyset$, select temperature $\tau$
2: **for** t = 1,...,T **do**
3:     Observe the data set $\mathcal{D}_t$
4:     $\mathcal{M}_t \leftarrow$ add random samples from $\mathcal{D}_t$ to the memory.
5:     **for** k = 1,...,$n_{iter}$ **do**
6:         Sample $\mathcal{D} = \{\boldsymbol{x}^{(n)}, y^{(n)}\}_{n=1}^N$ a batch from training data $\mathcal{D}_t$
7:         Sample an index i from $[0,\ldots,t-1]$

8: 
$$g_{\boldsymbol{\theta}} \leftarrow \nabla_{\boldsymbol{\theta}} \left[ \sum_{\boldsymbol{x},y\in\mathcal{D}} \mathcal{L}_t(\boldsymbol{\theta}, (\boldsymbol{x}, y)) + \sum_{\boldsymbol{x}\in\mathcal{M}_i} \lambda_i KL\left( \frac{1}{\tau} f_{\boldsymbol{\theta}_i^*}(\boldsymbol{x}) \,\middle\|\, \frac{1}{\tau} f_{\boldsymbol{\theta}}(\boldsymbol{x}) \right) \right]$$

9:         $\boldsymbol{\theta} \leftarrow \boldsymbol{\theta} - \alpha \cdot \text{SGD}(\boldsymbol{\theta}, g_{\boldsymbol{\theta}})$
10: **return** $\boldsymbol{\theta}$

---

## 4 RELATED WORK

Continual learning has been studied in different fields of AI such as robotics (Thrun & Mitchell, 1995), computer vision (Li & Hoiem, 2017) and machine learning (Kirkpatrick et al., 2017). In this work, we focus to study continual learning mainly from the machine learning perspective.

The goal of continual learning is to build a learner that can continuously learn from a stream of tasks while still maintain its previously acquired knowledge. Prior works can be broadly categorized into

4 main approaches: (1) structural regularization, (2) functional regularization, (3) Bayesian learning and (4) ensemble learning.

**Structural regularization** approaches employ penalties on the network's parameters, encourage the important parameters of previous tasks to not change when learning the new task. The parameter importance can be estimated by calculating the Fisher information as in EWC (Kirkpatrick et al., 2017) or directly measure how changes in each parameter will affect the change in the loss function as proposed in SI (Zenke et al., 2017). Exact estimation of the full Fisher is intractable. Therefore, both EWC and SI estimates only its diagonal, namely, how much each parameter contribute to the total loss assuming that the parameters are independent from each other.

**Functional regularization** methods employ the regularizer on the model's output rather than its parameters. Different from structural regularization, functional regularization aims to preserve the learned input-output mapping function of the network. Learning without Forgetting (LwF) (Li & Hoiem, 2017) proposes to minimize the difference in KL-divergence of the current and previous models on the new tasks, which is a form of knowledge distillation penalty. Similarly, Less-forgetting learning (LF) (Jung et al.) proposes to minimize the $\ell_2$ norm of the two prediction in stead of KL-divergence as LwF. Motivated from Hebbian learning (Hebbs, 1949), MAS (Aljundi et al., 2018a) proposes to penalize the $\ell_2$ norm of the model's prediction. Selfless sequential learning (SNI) (Aljundi et al., 2018b) improves MAS by combining with it sparsity in each layer's activation. Different from LwF that distills previous models on the current data, which contradicts the goal of knowledge distillation since past models are not trained on the new data, they provide noise to the current model. Our approaches, DAR and SAR, distills past models on their correctly trained data, thus maintaining the past knowledge for new models.

**Bayesian learning** can be considered as a natural way of solving continual learning. Nguyen et al. (2018) proposes Variational Continual Learning (VCL) that using the posterior of all observed tasks as a prior to combine with the current log-likelihood to yield the new posterior, from which point can recurse. Huszár (2018) shows that from Bayesian learning, we can derive and simplify the objective of EWC into a single constraint. While there are some overlapping between Bayesian learning and our approximated multi-task learning framework, the main difference is that our objective allows us to actively control the contribution of previous tasks by setting the importance value $\lambda_i$ for each loss $\mathcal{L}_i$. In Bayesian learning, tasks importance are naturally assigned by the Bayes' formula, thus, further tasks might receive less importance compare to recently observed ones.

Finally, **ensemble learning** techniques addresses catastrophic forgetting by having a dedicated sub-network for each task. The learning of sub-networks can either be explicit or implicit. In explicit ensemble learning, each sub-network is either newly initialized whenever a new tasks arrives as in Progressive Neural Network (Rusu et al., 2016) or searched from a big network as proposed in Pathnet (Fernando et al., 2017). After training a task, the corresponding sub-network is frozen and will not allowed to change, thus completely immune to catastrophic forgetting. In contrast, implicit ensemble learning can be considered as a hybrid method of explicit ensemble with other types of regularization. Dynamically expandable networks (Yoon et al., 2018) proposes to learn sub-networks in a pre-intialized network through sparsity, adding new neurons whenever needed. Catastrophic forgetting is avoided by using a form of structural regularizer. Ensemble learning approaches are usually resistant to catastrophic forgetting as the cost of either unbounded growth of model's size or cannot be applied to different network architectures.

In this work, we propose an approximated multi-task learning framework to unify both structural and functional regularization, in which our proposed activity regularization is a special case.

## 5 EXPERIMENTS

### 5.1 EXPERIMENT SETTING

We evaluate the performance of DAR, SAR on three common benchmarks of continual learning: permuted MNIST, split notMNIST and split CIFAR-100. The baseline models are EWC (Kirkpatrick et al., 2017), SI (Zenke et al., 2017), GEM (Lopez-Paz et al., 2017) and VCL (Nguyen et al., 2018) (except CIFAR-100). Standard setting such as model architecture and data splits are used

Table 1: Average test accuracy of different methods at the end of training on 10 permuted MNIST. Asterisk denotes result is collected from the author's paper. Hyphen denotes default setting.

| Method | Memory type | Memory Size | Setting | Accuracy |
|---|---|---|---|---|
| EWC | model | 1.8m | $\lambda = 100, N = 200$ | 0.795 |
| SI | model | 0.2m | $c = 0.5$ | 0.860 |
| GEM | sample | 1.5m | - | 0.918 |
| VCL* | none | 0 | - | 0.900 |
| VCL + Coreset* | sample | 1.5m | - | 0.930 |
| **DAR** | **sample** | **1.5m** | $\lambda = 10, \tau = 5$ | **0.949** |
| **SAR** | **sample** | **1.5m** | $\lambda = 100, \tau = 5$ | **0.948** |

whenever possible. For DAR and SAR, we report the results with temperature $\tau = 5$ as we found it worked consistently good in all experiments.

Since the methods used in the experiments utilize different type of memory units: EWC and SI require storing the previous models and the parameter's importance estimation while GEM, VCL and our approaches use memory to store the training samples. We quantize the memory size used in each method by the number of floating point numbers used. For example, one MNIST image of size $28 \times 28$ requires 784 floating point numbers and a $100 \times 10$ matrix requires 1,000 float numbers.

## 5.2 PERMUTED MNIST BENCHMARK

Permuted MNIST is a popular benchmark for continual learning (Goodfellow et al., 2013; Kirkpatrick et al., 2017; Zenke et al., 2017). Here we use a series of 10 tasks each of which is generated by first randomly generate a permutation and then apply it to every image in the data set. The network architecture used in all experiments is a single-headed MLP with two hidden layers containing 100 neurons [784-100-100-10] with ReLU activation. The final layer is a single-head classifier with softmax outputs, that is, we only use one classifier for all 10 tasks.

For EWC, at the end of training, we randomly sample $N = 200$ samples to estimate the diagonal Fisher matrix and store the optimal model $\boldsymbol{\theta}_i^*$ for each task. For sample based memory approaches, we use random sampling to select $m = 200$ samples from each tasks and store them in the memory.

We perform hyper-parameters tuning on all approaches where we assume all previous tasks are equally important so that all $\lambda_i$ in eqn. 1 are equal. We experiment with $c \in \{0.5, 1\}$ for SI and $\lambda \in \{1, 10, 100, 300\}$ for other methods; we report the results according to the best value of these hyper-parameters. All models are optimized by SGD with learning rate of 0.05 over 10 epochs for each task; momentum and other forms of regularization are not used.

Table 1 reports the averaged test accuracies on all 10 tasks at the end of training. From the results, EWC perform the worst although requiring the most "memory unit". SI performs better than EWC thanks to its Fisher approximation strategy on the whole data. GEM attains 91.8% accuracy, higher than both EWC, SI and VCL due the its strong constraints. Combining VCL with coreset (analogy to our memory) improves its accuracy to 93%. Both DAR and SAR outperforms the baselines by large margins, achieving accuracy of 94.9% and 94.8% respectively.

We also investigate the scalability of our methods when more samples are available in the memory. In fig. 1, we plot the test accuracies after seeing 10 tasks of DAR, SAR and GEM. Both DAR and SAR are consistently better than GEM at all memory size. Only at 2500 samples, GEM performs better than our methods at 200 samples. At 2500 samples, both DAR and SAR achieves over 96% accuracy, showing great scalability. Across all methods, performance consistently improves when the memory size increases. When only a small amount of samples are available in the memory, e.g. 200 and 500, GEM may overfit to these samples, leading to the performance degrade on the whole past data sets. As more samples are stored, the memory can better represent the past tasks, results in performance improvement. In both cases, our methods show advantage over GEM, maintaining performance on previous task by enforcing model's consistency.

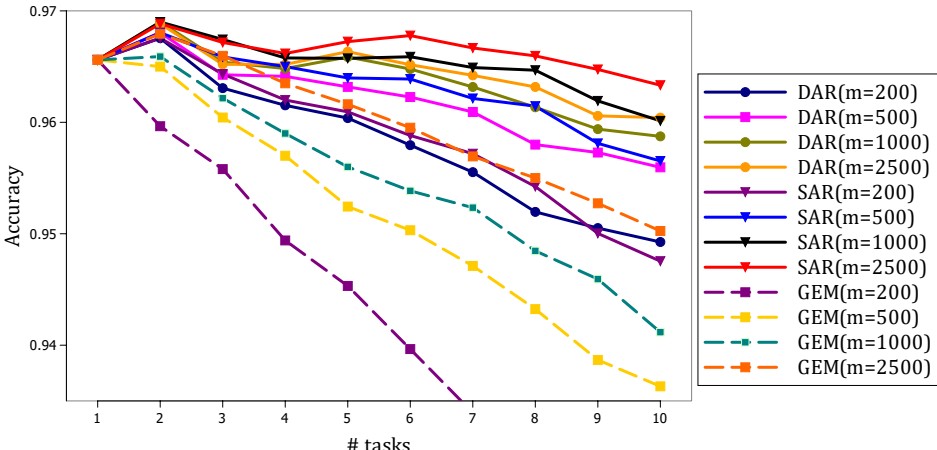

Figure 1: Comparison of different memory size on the permuted MNIST experiments.

Table 2: Average test accuracy on the split notMNIST benchmark.

| Method | EWC | GEM | SI | VCL* | VCL+Coreset* | **DAR** | **SAR** |
|---|---|---|---|---|---|---|---|
| Accuracy | 0.710 | 0.810 | 0.940 | 0.920 | 0.960 | **0.976** | **0.977** |

## 5.3 SPLIT NOTMNIST BENCHMARK

This experiment considers notMNIST[2], a more challenging version of MNIST. The notMNIST data set consists of over 500,000 images of characters A to J written in different fonts and has about 6.5% label error rate. Following Nguyen et al. (2018), we use the same data splits and consider five tasks: classifying the characters A/F, B/G, C/H and E/J. The training data consists of 400,000 images sampled from the original noisy 500,000 training data and the testing data is 18,000 cleaned images. We also use the same network architecture: a MLP with four hidden layers , each has 150 neurons [784-150-150-150-150-5×2]. The final layer is a multi-head classifier, that is, each task has a separated output. The other settings such as memory size, temperature are the same as the permuted MNIST experiment.

Table 2 reports the final averaged accuracy after learning five tasks. Compare to SI, EWC and GEM perform considerably worse due the presence of label noise. This result is consistent with our analysis in section 3.1: noisy memory can easily cause catastrophic forgetting in EWC and GEM while SI is more resistant to noise since it approximates the Fisher on all observed samples of past tasks. As a result, SI performs significantly better than both EWC, SI and VCL alone, achieving 94% accuracy. When combined with a coreset VCL further improves the performance to 96%. Finally, both DAR and VAR consistently perform better than the baselines, achieving 97.6% and 97.7% accuracy respectively.

## 5.4 SPLIT CIFAR-100 BENCHMARK

We consider the split CIFAR-100 benchmark (Zenke et al., 2017; Lopez-Paz et al., 2017), in which we split data CIFAR-100 data set into a series of 10-classes classification tasks. For this task, we use a CNN used in (Zenke et al., 2017), having 4 layers of convolutions, 2 fully connected layers with dropout in between. We compare DAR, SAR with EWC, SI and GEM and a finetuning model, VCL is not considered since it was not developed for convolutional layers. We use SGD optimizer with learning rate 0.05 over 30 epochs for all methods. The finetuning baseline is included in this experiments to measure how well we can learn without worrying about catastrophic forgetting. In previous experiments, the tasks are quite simple thus the network can easily achieve good performance.

---

[2] http://yaroslavvb.blogspot.com/2011/09/notmnist-dataset.html.

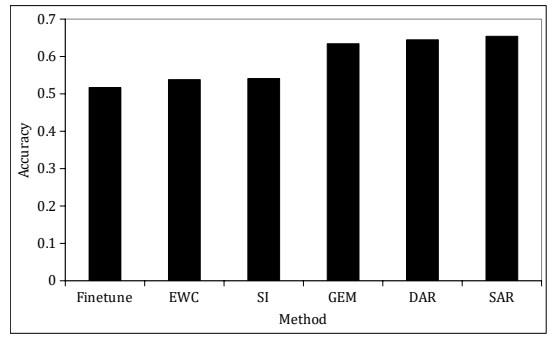
(a) Averaged accuracy on the split CIFAR-100.

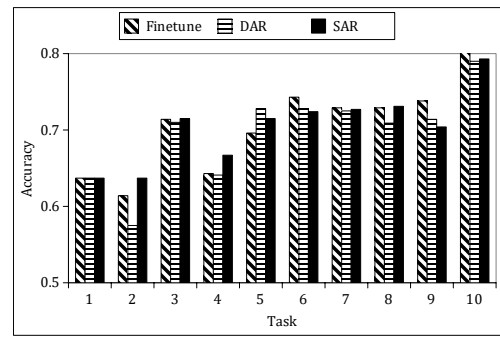
(b) Accuracy of each task on the split CIFAR-100.

Figure 2: Performance results on the split CIFAR-100 benchmark.

Fig. 2 shows the overall results of this experiments. We report the averaged accuracy after 10 tasks in fig. 2a, where EWC and SI are not as effective as alleviating forgetting as in previous experiments because this benchmark is more challenging. DAR and SAR achieve slightly higher averaged accuracy compare to GEM.

We also report individual task accuracy of DAR and SAR compare to the finetuning model in fig. 2. As expected, the finetuning model achieves relatively high performance on individual tasks because its goal is only to learn the current task by leveraging knowledge of previous tasks. Except the first task where all models are the same, SAR achieves slightly higher accuracy than DAR on 6 out of 9 remaining tasks. Overall SAR performs better than DAR by a small margin, showing that it can achieve similar results compare to DAR and has more scalability.

## 6 CONCLUSION

We propose an approximated multi-task learning framework that can alleviate catastrophic forgetting, an important challenge in continual learning and show that several previous solutions in the literature are special cases of this framework. We further analyze the weakness of two popular methods: EWC and GEM, showing that they are sensitive to mistakes and noise in the Fisher or memory. To overcome this problem, we propose the deterministic activity regularization for continual learning, DAR, that enforces the new model's behaviour to be consistent with the optimal models on each of the previous tasks. As the result, DAR is much more resistant to noise in the memory since its goal is to maintain the previously optimized decision boundary. We further improve DAR with a stochastic sampling regularizer (SAR), which balances the model's stability and plasticity, allowing the model to acquire new knowledge, alleviate forgetting and has greater scalability.

Through extensive experiments on popular continual learning benchmarks, we show that DAR and SAR consistently outperforms state-of-the-art methods. We demonstrate that label noise data such as notMNIST can be extremely destructive for EWC and GEM in the continual learning setting while our methods can still maintain good performance. Finally, we show that DAR and SAR can be applied on both MLP and CNN, effectively alleviate catastrophic forgetting.

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
