# OpenReview forum: "Activity Regularization for Continual Learning"
_ICLR.cc/2019/Conference_

### Official Review · AnonReviewer2 · 2018-11-02
**Simple approach, but limited novelty, and needs some improvement in exposition and benchmarking of related work**

**Rating:** 4
**Confidence:** 4

**Review:**

This paper proposes an approach to mitigate catastrophic forgetting in supervised learning by regularizing activations. The paper views previous techniques (EWC, SI, and GEM) under a multi-task learning lens, and then proposes an additional loss term to minimise the KL between activations from previous and current models, on previous tasks - this is based on a memory which stores some previous samples and their corresponding activations.

I think it is a simple and intuitive approach and a well-written paper. Unfortunately I have a number of concerns that I think preclude publication in the current state.

First, in terms of related work, I believe this is very similar to Learning without forgetting (LwF), with the difference that the KL-divergence is computed on samples kept from the previous tasks. This is briefly mentioned in the paper, but I think it needs to be made more explicit, and LwF should be a baseline in the experiments to clearly indicate the benefit of keeping this data. There is also a relationship to EWC: given the connection between the Fisher information and KL, it can be viewed as minimising the KL divergence in parameter space, rather than in activation space (which is the case here). Also note that EWC uses the true Fisher rather than the empirical, contrary to the derivation in equation (2).
There are also a number of papers that haven’t been cited in the related work [1][2][3][4].

Second, I think the motivation in Section 3.1 could be more convincing. Most importantly, it’s not clear to me that the decision boundary *shouldn’t* change for previously misclassified examples, as this could be an opportunity for backwards transfer.
Further, I don’t think the point in the last paragraph about having a small data portion is relevant, since they are from the same data distribution, and we would expect misclassified samples to be in the same (low) frequency in Fisher estimation as overall. I think the point of this paragraph is just that it is important to consider the entire predictive distribution of previous tasks rather than the probability of the correct class, so this should be stated more clearly and then justified.

Finally, I think the experimental justification could be improved as well. Beyond permuted MNIST (which it has been argued is not as useful as other baselines [4]), only the final performance on split notMNIST / CIFAR-100 is reported. Some comments and questions:
- The accuracies of EWC (and possibly SI) in the table are worse than reported in previous work (eg. [1]), so I think this needs to be examined.
- What is the fine-tuning baseline (I don't believe it is actually clearly defined)? How can it be so low in figure 2a but better in 2b?
- I think plots over time (performance on all tasks) would be much more useful than the final performance in Table 2 and Fig 2.
- Errors and error bars would be beneficial for all results.
- Table 1 should also include the references provided.

Some other comments and questions:
- Compared to eqn (2), eqn (6) is missing the ½ constant.
- Typos in section 5.3: "SI performs better than SI", and VAR instead of SAR.
- Section 2, unclear of meaning of "coined with the likelihood" (should this be “coincide”?)
- The first line should be “Humans have the ability to learn...” In general, I think the introduction could use another proofread for grammar and readability as I saw a few minor things.

[1] Nguyen, Cuong V., et al. "Variational Continual Learning." ICLR, 2018.
[2] Schwarz, Jonathan, et al. "Progress & Compress: A scalable framework for continual learning." ICML, 2018.
[3] Shin, Hanul, et al. "Continual learning with deep generative replay." NIPS, 2017.
[4] Farquhar, Sebastian, and Yarin Gal. "Towards Robust Evaluations of Continual Learning." arXiv, 2018.

---

### Official Review · AnonReviewer1 · 2018-11-02
**Interesting work but not good enough**

**Rating:** 4
**Confidence:** 5

**Review:**

The paper addresses the problem of continual learning from a sequence of supervised tasks. The main contribution of the paper are the following. The work:
* puts the problem in uniform general framework in which many of the state-of-the-art methods fit
* identifies some drawbacks of some the state of the art methods (namely EWC- Kirkpatrick et al., 2017 and GEM-Lopez-Paz et al., 2017) .
* proposes two versions of an approach to address these drawbacks
The main identified problem in EWC and GEM is that both methods result in a zero penalty when both the previous and current models misclassify a sample, even if they make different mistakes.
To solve this problem, the authors propose to keep a memory of randomly sampled data from the previous task, and distil the knowledge from the optima of the previous tasks to the current one using a KL penalty. The first version considers computing this penalty from the whole sequence of tasks, while the second randomly selects one task at each iteration and uses it to compute the penalty, paying some overall accuracy for plasticity.

While the paper is clear and well written, I have some concerns about it's quality, originality and significance.

Originality: The work seems to me to be very related to LwF. The main difference is that while LwF uses only the new data for the KL penalty, this paper keeps a memory of previously seen data to compute this loss.

Significance: While the way the authors approached the problem seems well structured and motivated, and is based on a sound observation, the authors limited themselves to experiments where the task data have similar structure. I am not sure how significant the improvement of the method would be in the more challenging and realistic setting where the data comes from different domains.  I am more specifically skeptical about the stochastic version of the algorithm in that case.

Quality:
* The proposed algorithm stores some data from previous tasks along with the outputs of the corresponding optimal model. While knowledge distillation as proposed would result in a non zero penalty when the new model makes different prediction, it still doesn't take advantage of the new information to probably correct the previous models prediction when it is wrong. Why not keeping the ground truth labels instead? This won't increase the memory requirement, and may give better results. It would be interesting to compare against such a method.
* I think selecting the samples to keep in memory randomly could to be suboptimal.  Other selection methods can be considered. A previous work:  iCaRL: Incremental Classifier and Representation Learning, Rebuffi et al. 2017, gives way to select representative samples. It would be interesting to see the effect of such a selection on the results.

Overall, while the paper doesn't present any significant flaw, it doesn't add much to the continual learning literature either, which explains my rating.

---

### Official Review · AnonReviewer3 · 2018-11-04
**The authors proposed a new regulariser for continual learning. However, the novelty is not clear and experiments setting needs improvement.**

**Rating:** 4
**Confidence:** 5

**Review:**

The authors proposed a new regularizer for continual learning to tackle the catastrophic forgetting problem. The proposed method minimizes the KL-divergence between the prediction of previous models and current models on the stored samples of previous tasks. The idea is straightforward and sounds technical. Experiments show the effectiveness of the methods compared to state-of-the-art. Although the idea sounds interesting and the experiments look promising, the novelty of the paper seems to be limited. In addition, the experiments setting needs to be improved as well. In the following, you have detailed comments.

1. It seems to be an extension of Learning without Forgetting (LwF) Li & Hoiem 2017 with simply on the examples of previous tasks in the memory. LwF only regularizes on the current task. It is not clear what is the difference between the proposed method and LwF except this.
2. The authors fixed many critical hyper-parameters: temperature(5), learning rate(0.05), epochs(10). The author should report the results for all methods with these hyper-parameters chosen on the validation set.
3. The authors presented how they split the training and testing data. Please be clear how you split the validation set.
4. The authors argued that sample quality does not affect the proposed methods. Then the authors should show the variance from different random sampling.

---

### Meta-Review · Area_Chair1 · 2018-12-13

**Confidence:** 5
**Recommendation:** Reject

**Metareview:**

There is no author response for this paper. The paper presents a multi-task learning framework as a unified view on the previous methods for tackling catastrophic forgetting in continual learning. In light of this framework, the authors propose to minimize the KL-divergence between the predictions of the previous optimal model and the current model using some stored samples from the previous tasks.

The consensus among all three reviewers and AC is that the paper lacks (1) novelty, as the proposed approach is similar if not identical to Learning without forgetting (LwF)[Li&Hoiem 2017] with the difference that the KL-divergence is computed on samples kept from the previous tasks (and LwF uses samples from the current task). Methodological and experimental comparison to LwF is crucial to assess the benefits and novelty of the proposed approach.

Also the reviewers address other potential weaknesses and give suggestions for improvement: (2) empirical evaluations can be substantially improved with sensitivity analysis of the hyper-parameters on the validation data (R3), indicating errors and error bars for all results (R3 and R2), using more challenging and realistic experimental setting where the data comes from different domains (R1), justifying the results better -- see R2’s questions; (3) lack of clarity and motivation in Section 3.1 -- see R2’s and R1’s suggestions for how to improve clarity and potentially take advantage of the current task to probably correct the previous models prediction when it was wrong.

AC suggests, in its current state the manuscript is not ready for a publication. We hope the reviews are useful for improving and revising the paper.